# Improving Black-box Robustness with In-Context Rewriting

**Kyle O'Brien**[1]     **Nathan Ng**[3,4,5]     **Isha Puri**[5]     **Jorge Mendez**[5]

**Hamid Palangi**[2]     **Yoon Kim**[5]     **Marzyeh Ghassemi**[5]     **Thomas Hartvigsen**[6]

[1]*EleutherAI*   [2]*Google*   [3]*University of Toronto*   [4]*Vector Institute*   [5]*MIT CSAIL*   [6]*University of Virginia*

**Reviewed on OpenReview:** *https://openreview.net/forum?id=e92dgUUfk0*

## Abstract

Machine learning models for text classification often excel on in-distribution (ID) data but struggle with unseen out-of-distribution (OOD) inputs. Most techniques for improving OOD robustness are not applicable to settings where the model is effectively a black box, such as when the weights are frozen, retraining is costly, or the model is leveraged via an API. Test-time augmentation (TTA) is a simple post-hoc technique for improving robustness that sidesteps black-box constraints by aggregating predictions across multiple augmentations of the test input. TTA has seen limited use in NLP due to the challenge of generating effective natural language augmentations. In this work, we propose LLM-TTA, which uses LLM-generated augmentations as TTA's augmentation function. LLM-TTA outperforms conventional augmentation functions across sentiment, toxicity, and news classification tasks for BERT and T5 models, with BERT's OOD robustness improving by an average of 4.48 percentage points without regressing average ID performance. We explore selectively augmenting inputs based on prediction entropy to reduce the rate of expensive LLM augmentations, allowing us to maintain performance gains while reducing the average number of generated augmentations by 57.74%. LLM-TTA is agnostic to the task model architecture, does not require OOD labels, and is effective across low and high-resource settings. We share our data[1], models[2], and code[3] for reproducibility.

## 1 Introduction

Text classification models deployed in real-world settings must excel on in-distribution (ID) inputs sampled from their training distribution and be robust to unseen out-of-distribution (OOD) inputs. OOD robustness is important for deploying safe and trustworthy models in real-world settings (Hendrycks et al., 2021). This challenge is especially acute in high-stakes settings such as content moderation (Ashish et al., 2023; Zhou et al., 2021), spam detection (Dada et al., 2019), and healthcare (Rasmy et al., 2020). OOD robustness in NLP is challenging in practice due to the complex nature of natural language data, adversarial examples (Goyal et al., 2022), and shifting domains (Yuan et al., 2023; Koh et al., 2020; Yang et al., 2023).

Most existing methods for improving OOD robustness in NLP require access to model weights by modifying the training process (Howard & Ruder, 2018; Ma et al., 2019; Ruder et al., 2019; Tu et al., 2020; Yaghoobzadeh et al., 2021) or adapting the model to new domains at test time (Wang et al., 2023b). Modifying the task model can be challenging in practice when retraining is costly, the underlying model is abstracted away from the practitioner, or sufficient OOD labels are unavailable. These constraints render the task model effectively a black box. In this work, we study the NLP task of short-form text classification in a black-box setting by turning attention towards the inputs to the model.

Test-time augmentation (TTA) sidesteps the need for modifying the task model or new labels by aggregating multiple predictions over augmentations of the test input, thus arriving at more robust predictions.

---

[1]https://huggingface.co/datasets/Kyle1668/LLM-TTA-Augmentation-Logs
[2]https://huggingface.co/collections/Kyle1668/
[3]https://github.com/Kyle1668/LLM-TTA

The choice of textual augmentation function is critical since augmentations must be diverse and semantic-preserving (Shanmugam et al., 2020), a challenge with conventional augmentation functions such as word insertion and synonym substitution (Xiong et al., 2023). LLM-driven advances in machine translation (Kocmi & Federmann, 2023; Wang et al., 2023a; Zhu et al., 2023), paraphrasing (Witteveen & Andrews, 2019; Wahle et al., 2022; Cegin et al., 2023), and style transfer (Patel et al., 2022; Suzgun et al., 2022; Roy et al., 2023) indicate that higher-quality augmentations are now feasible. In this work, we study using LLMs as the augmentation function for TTA (LLM-TTA).

Our study is composed of nine public datasets and one novel synthetic dataset across sentiment analysis, toxicity detection, and new topic classification. We consider the dataset used for model optimization as ID, while the other challenging evaluation sets are considered OOD. The experimental setup and its limitations are discussed in Sections 4.1 and 6.

We experiment with two LLM-TTA methods: zero-shot paraphrasing, where we prompt the LLM to generate paraphrases of the input text, and In-Context Rewriting (ICR), where the LLM rewrites the input to be more like a set of ID exemplars provided in the prompt. Both methods outperform TTA with conventional augmentation functions for BERT and T5 across averaged across nine datasets for sentiment, toxicity, and news topic classification. These results demonstrate that LLM-TTA is a simple black-box robustness technique effective across multiple tasks. Our primary findings are:

1. **LLM-TTA Improves OOD Robustness.** ICR improves a BERT classifier's absolute accuracy on OOD data by an average of 5.12% for sentiment, 7.18% for toxicity, and 1.15% for news topics, all with minimal regression to ID performance. For toxicity, BERT's ID accuracy improves by 2.99%, suggesting that LLM-TTA can improve both ID and OOD performance in some settings. Ablating training set size (Section 5.5) shows that LLM-TTA is useful in data-scarce and rich settings.

2. **TTA with Conventional Augmentation Functions Often Hurts Performance.** In contrast to LLM-TTA, TTA with conventional augmentation functions generally hurt both ID and OOD performance. Back-translation is the best-performing conventional augmentation functions, with BERT's OOD robustness improving by an average of 2.85 percentage points while word insertion regresses performance by –0.29 points and substitution by –0.53 points averaged across tasks.

3. **Selectively Augmenting High-Entropy Test Inputs Improves Efficiency.** We can reduce the rate of expensive LLM augmentations by only augmenting test inputs in which the task model is uncertain in its prediction. For BERT as the task model and ICR as the augmentation function, we reduce the percentage of test inputs requiring augmentation by an average of 57.74% while still improving robustness. The uncertainty threshold is only determined from ID statistics.

## 2 Related Work

**Text Augmentation.** Data augmentation is a strategy that enables practitioners to significantly increase the diversity and quantity of data available without collecting additional annotations (Feng et al., 2021). Train-time augmentation has been found to improve performance in low-resource scenarios (Chen et al., 2021), mitigating harmful features such as gender bias (Zmigrod et al., 2019), and improving robustness (Morris et al., 2020; Ng et al., 2020).

Textual data augmentation can be performed at the character (Karpukhin et al., 2019), word (Zhang et al., 2015; Kobayashi, 2018; Wei & Zou, 2019), or whole-text (Vickrey & Koller, 2008; Hou et al., 2018; Yu et al., 2018; Anaby-Tavor et al., 2019; Kumar et al., 2020) level. One concrete example is word substitution (Wei & Zou, 2019), where each word in the text has some probability of being replaced with a related word. Increasing the likelihood of replacement can result in more diverse augmentations but comes with the risk of losing the original semantic meaning of the source example (Xie et al., 2019; Bayer et al., 2021). Conversely, augmentations that do not introduce sufficient diversity are unlikely to be effective for large pretrained models (Longpre et al., 2020).

**Test-Time Adaptation.** This approach extends TTA by updating the weights of the source model at test-time (Liang et al., 2023). Adaptation is commonly implemented in an unsupervised manner by minimizing a proxy for supervised loss such as entropy (Wang et al., 2021; Zhang et al., 2021), distribution alignment

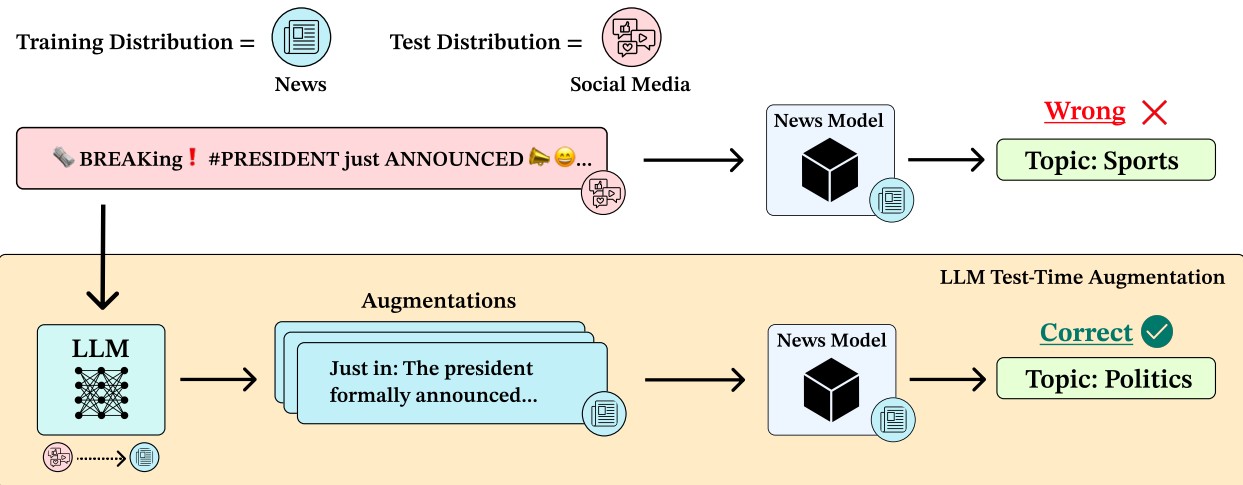

Figure 1: **LLM-TTA.** In settings where the task model is effectively a black box, we can intervene on the input data to improve robustness. We propose rewriting OOD inputs at test time using an LLM to improve robustness (LLM-TTA). Our experiments find that LLM-TTA improves performance without requiring task model access or OOD labels.

(Hassan et al., 2023), or through perturbing model internals such as with dropout (Liang et al., 2023). Crucially, this technique assumes the ability to modify the source model. Singh & Ortega (2022) extended the NLP TTA evaluation from Lu et al. (2022) to include adaptation and found that entropy minimization-based adaptation can improve performance accuracy on OOD toxicity detection.

**Test-Time Augmentation.**  TTA refers to the practice of aggregating predictions across multiple augmentations of an original test input during inference. This technique has been widely used in computer vision to improve ID performance (Moshkov et al., 2019; Ashukha et al., 2021), OOD robustness (Enomoto et al., 2022; Kim et al., 2020; Molchanov et al., 2020), adversarial robustness (Song et al., 2017; Prakash et al., 2018; Cohen et al., 2019), and uncertainty estimation (Ayhan & Berens, 2018; Gawlikowski et al., 2021). Lu et al. (2022) provided the initial foray into studying TTA for NLP. They studied how common word-based augmentations improve the robustness of a distilBERT (Sanh et al., 2019) on the WILDS CivilComments (Koh et al., 2020) dataset. Matiana et al. (2021) combines cosine similarities of model representations across multiple paraphrases of the input to classify stories. Xiong et al. (2023) explored dynamically assigning different weights to augmentations during aggregation to reduce the influence of potentially noisy augmentations. These works found that incremental improvements in accuracy are possible but that the word-based augmentation functions are a bottleneck for performance. Our study differs from these works in that we focus our evaluation on LLM-based TTA and across a more diverse set of tasks, models, and augmentation functions, as well as measure ID performance alongside OOD robustness.

## 3   LLM-TTA: Generating Faithful Augmentations With LLMs

In this section, we describe LLM-TTA, a method for performing test-time augmentation for natural language classification using the rewriting capabilities of large language models.

### 3.1   Augmenting Test Inputs

In this paper, we consider a supervised classification task from an input space $\mathbf{X} \in \mathcal{R}^n$ to a discrete output space $\mathbf{Y}$ with $K$ classes. We assume access to a model $f$ trained on a dataset $\mathcal{D}_{train} = \{(\mathbf{x}_i, \mathbf{y}_i)\}_{i=1}^n$ sampled from a distribution $p_{train}(\mathbf{X}, \mathbf{Y})$. At test-time, we are given a point $\mathbf{x}$ sampled from an unknown distribution $p_{ood}(\mathbf{X}, \mathbf{Y})$, which may or may not be the same as our original training distribution.

TTA is a simple post-training approach for improving the generalization accuracy of $f$ on arbitrary inputs $\mathbf{x}$. The main aim of TTA is to reduce the effects of a poor prediction on a single point by aggregating them across many similar augmented points. TTA involves three steps: augmentation, inferences, and aggregation.

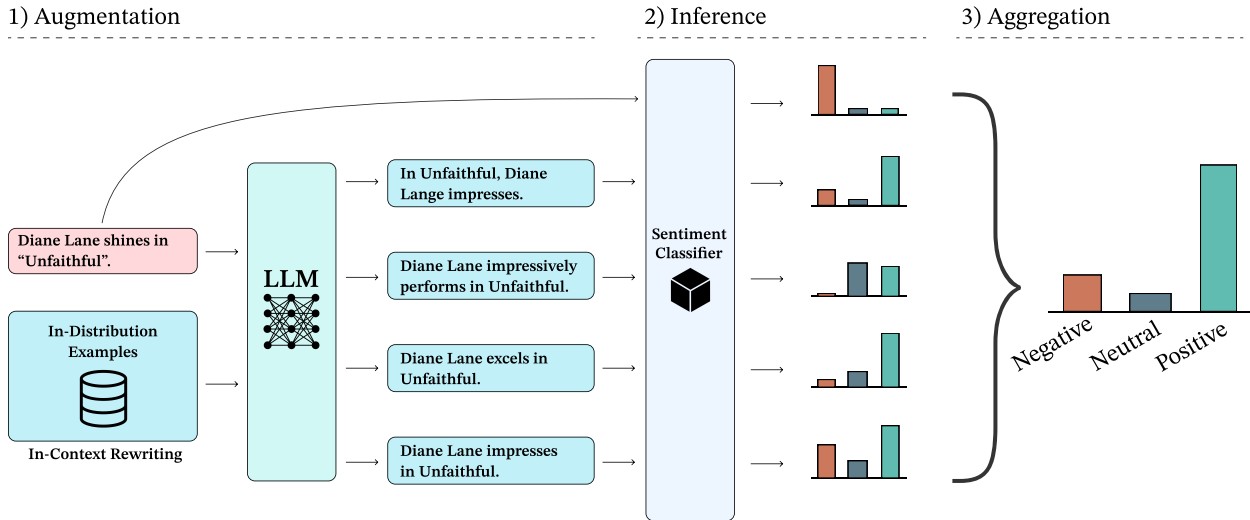

Figure 2: **TTA Inference Steps.** This figure shows the three stages of TTA. The process begins with an augmentation function generating multiple altered versions of the current test input. For ICR, the input to the LLM also contains ID examples. The task model then makes predictions over the test input and its augmentation. Lastly, we aggregate the predictions to arrive at a "smoothed" judgment. Standard aggregation methods include mean probability aggregation (demonstrated in this figure) and vote-based aggregation.

As described in Section 2, TTA differs from test-time adaptation (Wang et al., 2023b) in that the weights of the source model remain frozen. Unlike train-time augmentation (Shorten & Khoshgoftaar, 2019; Bayer et al., 2021), TTA does not require practitioners to modify their training regimes. Section 2 provides further description of these alternative techniques. We use terminology inspired by Shanmugam et al. (2020) and notation from Goodfellow et al. (2016).

1. **Augmentation.** We define an augmentation $a_i$ as a stochastic transformation of an input $\mathbf{x}$ from which we can sample another augmented point $\mathbf{x}' \sim a_i(\mathbf{x}'|\mathbf{x})$. Given a set of such augmentations $M = \{a_i\}_{i=1}^m$, we define $M(\mathbf{x}) = \{\mathbf{x}'_i \sim a_i(\mathbf{x}'|\mathbf{x})\}_{i=1}^m$ as a set of single samples from each of the augmentations in $M$. For ease of notation, we assume one of the transformations in $M$ is the identity transformation $I(\mathbf{x}) = \mathbf{x}$ such that the original point $\mathbf{x}$ is always in $M(\mathbf{x})$.

2. **Inference.** Each of the points in $M(\mathbf{x})$ is passed into the model $f$ to generate a set of predictions $f(M(\mathbf{x})) = \{f(\mathbf{x}'_i)\}_{i=1}^m$

3. **Aggregation.** A final prediction $\hat{\mathbf{y}} = G(f(M(\mathbf{x}))$ is derived from an aggregation function $G$ that combines the set of predictions $f(M(\mathbf{x}))$. Common aggregation methods include vote-based aggregation, where the most commonly predicted class is chosen, and mean-based aggregation, where the probabilities for each class are first averaged across augmented samples before a prediction is made.

## 3.2 Rewriting Test Inputs with Language Models

We study whether employing TTA with augmentations generated by an LLM (LLM-TTA) will improve a task-specific classifier's robustness. LLMs have achieved strong performance on adjacent tasks that require faithfully rephrasing a target text. We hypothesize that the LLM-generated augmentations will outperform conventional augmentation functions. If so, these results suggest that LLM-TTA is a promising general non-parametric technique that NLP practitioners can leverage without depending on the task model architecture or access to weights.

In LLM-TTA, $\mathbf{x}' \sim \text{LLM}(\mathbf{x}'|P, \mathbf{x})$ is an inference process where an augmentation of the natural language test input $\mathbf{x}$ is generated by a language model (LLM) conditioned with a natural language prompt $P$ as well as the original input $\mathbf{x}$. We study two prompt templates for $P$ as detailed in Figure 3.

```
LLM-TTA: Paraphrasing

### Instructions ###
The assistant is to paraphrase
the input text.

### Input Text ###
Now paraphrase {{{"<style_input>"}}}.

Return the text in the
format: {{{Paraphased Text}}}.

### Paraphrased Text ###
Paraphrased Input Text: {{{### Assistant:
```

```
LLM-TTA: In-Context Rewriting (ICR)

### Instructions ###
The assistant is to paraphrase the input
text as if it was one of the examples.
Change the details of the text
if necessary.

### Style Examples ###
<style_transfer_exemplars>

### Input Text ###
Now paraphrase ‘‘‘"<style_input>"’‘‘‘ as
if it was one of the examples. Change
the details of the text if necessary.

Return the text in the
format: ‘‘‘Paraphrased Text‘‘‘

### Paraphrased Text ###
Paraphrased Text:
```

Figure 3: **LLM-TTA Prompts.** We evaluate LLM-TTA with two prompting methods. *"<style_input>"* is replaced with the test input and *"<style_transfer_exemplars>"* with the ID examples during inference. During the course of prompt engineering, we find that instructing the LLM to generate text in specific formats surrounded by brackets and to change the details of the text while preserving semantics leads to the best performance.

1. **Paraphrasing.** $P$ contains instructions for the model to generate a semantic-preserving paraphrase of the test input. This approach is zero-shot in that $P$ contains no examples, thus not requiring access to ID data. We expect simple paraphrasing done by a capable LLM can generate diverse and semantic preserving augmentations, even for subtle labels such as sentiment.

2. **In-Context Rewriting (ICR).** In this approach, we prompt the LLM to rewrite the given OOD example to be more like a set of ID exemplars. ICR does not require practitioners to articulate the semantics or writing style of the original distribution. Instead, the LLM is prompted to infer the differences between examples and $x$ via in-context learning. We hypothesize that rewriting OOD inputs to be more like ID data can further improve performance over simple paraphrasing since the task model was trained to excel on ID data.

### 3.3 Entropy-Based Selective Augmentation

LLM inference is inefficient when the model's predictions are invariant across the test input and its augmentations. This invariance is a blocker for overruling an incorrect prediction for the test input. To reduce the rate of expensive LLM inferences, we explore whether the entropy of the task model's class probability distribution is a predictive heuristic for whether the model will make an incorrect prediction. We can selectively augment test examples that are likely to benefit from TTA while deferring to the model's original prediction otherwise. Lower entropy has been observed to be correlated with correct predictions in machine learning models (Grandvalet & Bengio, 2004). This observation has motivated entropy to be leveraged as an unsupervised loss metric in test-time adaptation techniques (Wang et al., 2021; Zhang et al., 2021; Singh & Ortega, 2022).

Entropy-based selective augmentation (ESA) can be formally defined as follows. Let $\mathbf{x}$ be the test input, $H_f(\mathbf{x})$ be the entropy of the output probability distribution $p_f(\mathbf{y}|\mathbf{x})$ of the model $f$, and $e$ be an entropy threshold. For ease of notation, let $TTA(\mathbf{x}; f; M)$ be the aggregated prediction for $f$ using TTA with an augmentation function $M$. We then define the model prediction $\hat{\mathbf{y}}$ as

$$\hat{\mathbf{y}} = \begin{cases} \text{TTA}(\mathbf{x}; f; M) & \text{if} \quad H(\mathbf{x}) \geq e \\ \arg\max_{\mathbf{y}} p_f(\mathbf{y}|\mathbf{x}) & \text{otherwise} \end{cases} \tag{1}$$

TTA is only used when the original prediction entropy is above a predetermined threshold. The percentage of examples above this threshold (thus requiring augmentation) is referred to as the *augmentation rate*. We assume access to the labels for the ID evaluation set to determine an optimal threshold. Within this evaluation set, an optimal threshold is calculated that balances gains in accuracy while minimizing the augmentation rate, details of which are described in Appendix A.4.

## 4 Experimental Setup

We study how well LLM-TTA can improve the robustness of various task models across three short-form text classification tasks. We simulate a black-box setting by keeping the weights of the task model frozen at test time and study methods that are task model architecture independent. The following sections describe our datasets, task models, and TTA augmentation functions.

### 4.1 Datasets

We use three ID evaluation datasets and seven OOD datasets across sentiment analysis, toxicity detection, and news topic classification. Each task model is optimized for the ID evaluation set either through fine-tuning for BERT and T5, or prompting with Falcon. The training splits for the OOD datasets are not used. Dataset statistics are listed in Appendix A.6.

We refer to the dataset that we optimize the model for as ID. The challenging splits from other datasets, which the ID model struggles to generalize, are considered OOD. We do not investigate the specific properties that make these datasets challenging. This abstract notion of OOD contrasts with other works (Hendrycks et al., 2020a; Koh et al., 2020). See Section 6 for further discussion.

**Sentiment Classification.** We consider three sentiment classification datasets from the BOSS benchmark (Yuan et al., 2023). Each is a three-way classification task where the labels are positive, neutral, and negative. We keep this benchmark's selection of ID and OOD splits. The ID dataset consists of Amazon reviews (McAuley & Leskovec, 2013), while the three OOD datasets are *DynaSent* (Potts et al., 2020), *SST-5* (Socher et al., 2013), and *SemEval* (Nakov et al., 2016). Yuan et al. (2023) selected these three OOD shifts since their centroids had low cosine similarity with the ID (Amazon) centroid.

**Toxicity Detection.** We also leverage the toxicity task in the BOSS benchmark. Toxicity detection is framed as a binary classification task between non-toxic (negative) and toxic (positive). We similarly rely on the ID and OOD dataset selections from Yuan et al. (2023). The ID dataset is *Civil Comments* (Borkan et al., 2019), a collection of users' comments on articles on the Civil News platform. The OOD datasets are *AdvCivil*, an adversarial version of Civil Comments introduced in the benchmark, as well as existing datasets *ToxiGen* (Hartvigsen et al., 2022) and *Implicit Hate* (Elsherief et al., 2021). As with sentiment, these shifts were selected by Yuan et al. (2023) since they have low cosine similarity with the ID centroid.

**News Topic Classification.** AG News (Zhang et al., 2015) is a four-class news topic classification problem where the model is tasked with determining whether a given news article pertains to "World," "Sports," "Business," or "Sci/Tech." This task has a single OOD dataset, a novel dataset that is composed of the AG News test set style-transferred to resemble social media posts. Since each entry in the ID evaluation set has a corresponding style transferred entry in the OOD set, we can isolate the effect that differences in writing style have on performance. Additional details on how this dataset was created can be found in Appendix A.7.

### 4.2 Task Models

We study black-box robustness for three task models. First, we fine-tune BERT (Devlin et al., 2019) and T5-Large (Raffel et al., 2020) for each ID dataset. Second, we consider Falcon-7b, an instruction-tuned LLM (Almazrouei et al., 2023). To make predictions with Falcon, we use 16-shot prompt with randomly selected

ID exemplars with equal numbers per class. Fine-tune details are in Appendix A.5, and Falcon prompting details are in Appendix A.9.

### 4.3 Baseline Augnmention Functions

We compare our LLM-TTA augmentation functions with three representative functions studied in the NLP literature.

**Word-Level Augmentations.** Existing works studying TTA for NLP have focused on world-level augmentations (Lu et al., 2022; Xiong et al., 2023). We use the word insertion and substitution methods implemented in *nlpaug* recommended by Lu et al. (2022). We use this library's default parameters, where each word in the text has a 30% chance of being augmented for a maximum of 10 words. Insertion adds a new word after a target word in the text based on the word BERT predicts will come next based on the preceding and subsequent words. Substitution follows a similar approach, with the target word being replaced. Using BERT's predictions reduces the chance of adding nonsensical words that may change the semantics of the text.

**Back-Translation.** We select English↔German back-translation as a representative whole-text augmentation function. While not studied in the NLP TTA literature, translation is a common data augmentation technique (Edunov et al., 2018; Xie et al., 2019; Ng et al., 2019). Stochastic translations act as paraphrases of the original text. We select German as the target language since English↔German is a common pairing in the literature (Edunov et al., 2018; Sennrich et al., 2015; Hoang et al., 2018). Including a whole-text augmentation function in our experiments allows us to study how well paraphrasing with and without an LLM affects performance. Model and decoding specifics are included in Appendix A.9.

### 4.4 Multiple Evaluation Runs

To ensure that performance is robust across generated augmentation, we run our OOD evals four times using various seeds: 3, 16, 46, and 58. The results reported for OOD shifts are the mean values derived from these four runs, as well as the standard deviation. Each augmentation function is non-deterministic, which results in differing augmentations being generated across runs. This approach allows us to assess how consistently each augmentation function affects performance. We conduct a single run for the ID datasets.

### 4.5 TTA Settings

**Aggregation.** Following Lu et al. (2022), each experiment uses the test input and four augmentations to arrive at a final prediction. We only use a single augmentation function per experiment rather than mixing augmentation functions to better study the effect that specific augmentation functions have on performance. We use two different aggregation methods for mapping the predictions over the augmentations to a final class prediction. For BERT, we average the class probability distribution across the test input and its four augmentations and select the class with the highest probability. T5 and Falcon use a vote-based aggregation method where a verbalizer function maps a set of task-specific valid tokens (such as "1", "pos," and "positive") to a class label. We then select the most commonly predicted class across the five inputs as the final prediction.

**LLM-TTA.** We use Stable Beluga 2-7B (SB2) to generate augmentations. SB2 is a LLama 2 (Touvron et al., 2023) model fine-tuned with additional instruction tuning. Figure 3 details the prompts for the LLM-TTA paraphrasing and ICR methods. These prompts and the test input are passed to the model, resulting in four stochastic augmentations. ICR uses 16 randomly selected unlabeled exemplars balanced across classes sourced from the ID training set. We further describe the decoding details in Appendix A.9.

**ESA Threshold.** As introduced in Section 3.3, we study whether only augmenting test inputs in which the predicted class probability distribution is above a predetermined threshold still improves performance while reducing the augmentation rate. We find the optimal threshold for the ID test set that strikes a balance between performance gains and augmentation rate. We do not rely on access to OOD labels. We further describe this methodology in Appendix A.4.

| Augmentation | Sentiment | | | Toxicity | | | News → Tweets | | |
|---|---|---|---|---|---|---|---|---|---|
| | BERT | T5 | Falcon | BERT | T5 | Falcon | BERT | T5 | Falcon |
| None | 52.05% | 58.09% | **46.77%** | 53.88% | 58.83% | **59.11%** | 88.57% | 88.99% | 25.86% |
| Insert (Lu et al., 2022) | 51.66% $_{\pm 0.5}$ | 56.11% $_{\pm 0.2}$ | 45.82% $_{\pm 0.3}$ | 53.11% $_{\pm 0.4}$ | 57.59% $_{\pm 0.3}$ | 57.33% $_{\pm 0.1}$ | 88.86% $_{\pm 0.1}$ | 89.69% $_{\pm 0.1}$ | 24.98% $_{\pm 0.1}$ |
| Substitute (Lu et al., 2022) | 51.44% $_{\pm 0.3}$ | 54.15% $_{\pm 0.5}$ | 43.56% $_{\pm 0.3}$ | 52.70% $_{\pm 0.3}$ | 56.99% $_{\pm 0.5}$ | 58.87% $_{\pm 0.1}$ | 88.77% $_{\pm 0.1}$ | 89.08% $_{\pm 0.1}$ | 24.78% $_{\pm 0.1}$ |
| Translate (Sennrich et al., 2015) | 52.41% $_{\pm 0.1}$ | 56.60% $_{\pm 0.0}$ | 44.02% $_{\pm 0.0}$ | 62.11% $_{\pm 0.3}$ | 63.40% $_{\pm 0.0}$ | 59.08% $_{\pm 0.0}$ | 88.53% $_{\pm 0.1}$ | 89.13% $_{\pm 0.0}$ | 26.88% $_{\pm 0.0}$ |
| LLM–TTA: Paraphrase | 55.24% $_{\pm 0.2}$ | 59.01% $_{\pm 0.3}$ | 45.15% $_{\pm 0.2}$ | **64.67%** $_{\pm 0.3}$ | **67.20%** $_{\pm 0.5}$ | 58.90% $_{\pm 0.1}$ | 88.97% $_{\pm 0.0}$ | 89.98% $_{\pm 0.1}$ | 27.50% $_{\pm 0.2}$ |
| LLM–TTA: ICR | **57.17%** $_{\pm 0.2}$ | **59.75%** $_{\pm 0.2}$ | 43.79% $_{\pm 0.1}$ | 61.06% $_{\pm 0.2}$ | 63.83% $_{\pm 0.3}$ | 58.11% $_{\pm 0.1}$ | **89.72%** $_{\pm 0.1}$ | **90.30%** $_{\pm 0.1}$ | **27.82%** $_{\pm 0.2}$ |

Table 1: **OOD TTA Performance.** We report the mean and standard deviation of task model accuracy with TTA using various augmentation functions across four runs. Sentiment and toxicity results are averaged across the three OOD shifts for each task. Results are divided between TTA with conventional augmentation functions and LLM-TTA (our method). LLM-TTA is the best-performing augmentation function for BERT and T5 across all tasks. Larger models tend to benefit less from TTA, with all TTA methods hurting Falcon's sentiment and toxicity performance. These results suggest that LLM-TTA can be a useful technique for improving task model robustness.

| Augmentation | Sentiment | | | Toxicity | | | News → Tweets | | |
|---|---|---|---|---|---|---|---|---|---|
| | BERT | T5 | Falcon | BERT | T5 | Falcon | BERT | T5 | Falcon |
| None | 90.38% | 90.12% | **90.49%** | 88.46% | 90.57% | 17.01% | 94.43% | **94.87%** | **30.25%** |
| Insert (Lu et al., 2022) | 90.48% | 88.40% | 89.41% | 89.25% | 90.92% | 16.94% | **94.89%** | 92.22% | 26.84% |
| Substitute (Lu et al., 2022) | 89.96% | 85.78% | 86.58% | 89.90% | 91.33% | 19.99% | 94.86% | 91.91% | 26.34% |
| Translate (Sennrich et al., 2015) | 89.05% | 87.22% | 87.14% | 89.64% | 90.65% | 17.39% | 93.92% | 92.70% | 30.12% |
| LLM–TTA: Paraphrase | 90.54% | 89.90% | 90.35% | 90.58% | 91.32% | 18.43% | 93.72% | 92.05% | 28.78% |
| LLM–TTA: ICR | **90.85%** | **90.78%** | 90.35% | **91.45%** | **91.87%** | **19.00%** | 93.53% | 91.99% | 29.47% |

Table 2: **ID TTA Performance.** We continue the format from Table 1 to report ID accuracy. Results are from a single experiment run. While TTA works best as an OOD robustness technique, it can also improve ID performance. LLM-TTA improves BERT and T5 performance on sentiment and toxicity yet regresses performance on news. Similar to OOD robustness, TTA is most effective for BERT and can often hurt Facon's performance. These results suggest that LLM-TTA can improve OOD robustness without regressing ID performance, but the degree of gains depends on the model and task.

## 5 Results

### 5.1 LLM-TTA Improves OOD Robustness

Generating augmentations with LLMs outperforms all other augmentation functions for BERT and T5 across each OOD task, as shown in Table 1. We report the average accuracy across the multiple OOD shifts for sentiment and toxicity which is then average across four experiment runs. ICR is the best-performing augmentation function, with BERT's OOD performance improving by an average of 4.48 percentage points and T5's by 2.66 percentage points. Falcon benefits less with LLM-TTA regressing performance on sentiment and toxicity for an overall net regression of –0.67 percentage points.

ICR is the overall best-performing augmentation function. Augmenting test inputs to be more like ID exemplars outperforms 0-shot paraphrasing. These gains are achieved without requiring the entire ID dataset at test time, OOD labels, or explicit descriptions of the original distribution within the prompt. 0-shot paraphrasing also outperforms all conventional augmentation functions and ICR for the toxicity task. These results suggest that LLM-generated augmentations generally outperform conventional augmentations, and that performance can be further improved for some tasks by leveraging ICR over simple paraphrasing.

LLM-TTA can improve task model robustness without modifying the task model's weights or training regime. TTA's flexibility allows it to be slotted into existing systems. The degree to which TTA improves robustness is contingent on the augmentation function, with whole-text augmentation functions outperforming word-level augmentations. The modest gains, even with a powerful augmentation function, suggest that augmentation quality may not be the only bottleneck for improving black-box robustness. While we observe improvements, further progress is needed to make models more robust to OOD shifts as test time.

| Augmentation | Sentiment | | Toxicity | | News → Tweets | |
|---|---|---|---|---|---|---|
| | ID | OOD | ID | OOD | ID | OOD |
| LLM-TTA ICR: BERT | **90.85%** | 57.17% | 91.45% | 61.06% | **93.53%** | 89.72% |
| LLM-TTA ICR: T5 | 90.78% | **59.75%** | **91.87%** | 63.83% | 91.99% | **90.30%** |
| LLM-TTA ICR: Falcon | 90.35% | 43.79% | 19.00% | 58.11% | 29.47% | 27.82% |
| LLM | 86.53% | 50.63% | 76.32% | **71.76%** | 81.39% | 80.50% |

Table 3: **Direct LLM Performance vs LLM-TTA.** We report ID accuracy and mean OOD accuracy across shifts and seeds for the augmentation LLM (SB2) and the task models with LLM-TTA. BERT and T5 outperform the LLM on all ID datasets and 2/3 OOD shifts. Falcon generally underperforms SB2. SB2 outperforms all task models in terms of OOD toxicity performance. These results suggest that LLM-TTA can be effective even when the LLM underperforms the task model, but there may be instances where using the LLM directly is more efficient.

| Augmentation | Sentiment | | Toxicity | | News → Tweets | |
|---|---|---|---|---|---|---|
| | Accuracy | Aug Rate | Accuracy | Aug Rate | Accuracy | Aug Rate |
| None | 52.05% | - | 53.88% | - | 88.57% | - |
| Paraphrase: Default | 55.24% $_{\pm 0.2}$ | 100.00% | 64.67% $_{\pm 0.3}$ | 100.00% | 88.97% $_{\pm 0.0}$ | 100.00% |
| Paraphrase: ESA | 54.77% $_{\pm 0.1}$ | 50.51% | 61.24% $_{\pm 0.3}$ | 66.50% | 88.80% $_{\pm 0.1}$ | 1.62% |
| ICR: Default | 57.17% $_{\pm 0.2}$ | 100.00% | 61.06% $_{\pm 0.2}$ | 100.00% | 89.72% $_{\pm 0.1}$ | 100.00% |
| ICR: ESA | 56.38% $_{\pm 0.2}$ | 56.53% | 58.33% $_{\pm 0.1}$ | 66.50% | 89.21% $_{\pm 0.0}$ | 3.76% |

Table 4: **BERT Performance with Entropy-Based Selective Augmentation (ESA).** Metrics are mean accuracy across runs and augmentation rate: the percentage of test inputs that are augmented. The augmentation rate is identical across runs. **Default** refers to when every test input is augmented. Selective augmentation can improve performance while augmenting significantly fewer inputs, thus reducing the cost of expensive LLM-based augmentation.

## 5.2 LLM-TTA Can Improve ID Performance

Preserving ID performance is essential in settings where the overall distribution has only partially shifted. Table 2 reports ID gains using TTA. LLM-TTA improves ID performance for BERT and T5 for sentiment and toxicity but not for news. ICR is the best-performing augmentation function for these models and distributions. All augmentation functions regress Falcon's ID performance on sentiment and news. ID results follow a similar pattern to the OOD results reported in Table 1, where LLM-TTA performs the best for BERT and T5. These results suggest that LLM-TTA does not seriously regress ID performance and can even improve ID performance in some settings.

## 5.3 LLM-TTA Can Outperform the LLM

LLM-TTA augments every test input by default. In practice, using the augmentation LLM directly for the task may be advantageous if it can outperform the task model. In this section, we evaluate Stable Beluga 2 (SB2), the LLM we use for augmentation in our experiments, directly on the tasks. SB2 is prompted with the same task templates and exemplars as Falcon detailed in Appendix A.9.

We report our results in Table 3. SB2 underperforms BERT on all ID datasets and most OOD sets. SB2 overall outperforms BERT and T5 on the OOD toxicity detection and outperforms Falcon across most tasks. These results show that LLM-TTA can still improve a task model's robustness even when the LLM itself underperforms the task model. However, using the LLM directly may be practical in some settings.

### 5.4 Selective Augmentation Improves Efficiency

Tables 1 and 2 demonstrate that LLM-TTA can improve performance when every test input is augmented. However, augmenting each input can be costly when using LLMs. We introduced entropy-based selective augmentation (ESA) in Section 3.3 to address this issue. ESA only augments test inputs in which the model is uncertain in its prediction. We use the entropy of the OOD test input's predicted class probability distribution as the confidence measure. The intuition behind this method is that TTA is unlikely to be effective when the task model is confident in its prediction for the unaugmented test inputs and is thus unlikely to change its prediction across faithful augmentations. We use BERT as our task model since this model explicitly outputs a class probability distribution and benefits the most from TTA.

Table 4 demonstrates that selective augmentation can preserve most performance gains while drastically reducing the number of expensive LLM calls. ICR tends to benefit more than paraphrasing. BERT's robustness on sentiment is improved by 4.33 points when only augmenting 56.53% of inputs compared to 5.12 points when augmenting every input. Similarly, the majority of gains are still realized for toxicity while reducing the augmentation rate by roughly a third. News only preserved half of the performance gains yet reduced the augmentation rate by over 95% for ICR and paraphrasing.

That performance can still be improved while augmenting drastically fewer OOD inputs suggests that TTA does not change most model predictions. Selective augmentation is a promising direction for narrowing in on test inputs likely to benefit from augmentation while avoiding those that will not benefit. The entropy-based approach we studied can meet this criterion without requiring OOD labels. Future techniques can improve upon this approach by leveraging other signals beyond entropy.

### 5.5 LLM-TTA Is Effective in Both Data Scarce & Rich Settings

The BERT models studied previously are trained on the full training set numbers tens of thousands of examples. Whether LLM-TTA is effective across data scales is of interest since many practitioners operate in data-scarce regimes. It is unclear if LLM-TTA's performance generalizes to models trained on far fewer examples.

In this experiment, we study whether LLM-TTA can improve task model robustness across data scales. We train 5 BERT models on 5%, 10%, 20%, 40%, and 80% of the ID training set for each of our three tasks. The base models and hyperparameters are identical across runs and follow the training regime outlined in appendix section Appendix A.5. We build each balanced training subset via stratified random sampling across classes.

Figure 4 shows the average performance improvement over the no-TTA baseline averaged across OOD shifts. LLM-TTA's performance peaks at 10% of the training set for sentiment, 100% for toxicity, and 20% for news. The deltas between the best-performing dataset size and the fully trained sets are generally small, suggesting that TTA is only marginally more useful in low-resource settings. The takeaway for practitioners is that, given identical base models, LLM-TTA's performance in high-resource settings broadly generalizes to low-resource settings and vice versa.

### 5.6 LLM-TTA Affects Some Classes More Than Others

Whether TTA affects some classes more than others is of interest since many ML techniques can increase net performance while hurting specific classes or subgroups (Shanmugam et al., 2020; Yang et al., 2023; Kirichenko et al., 2023). Figure 5 shows the percent of all changed judgments broken down by class for BERT using ICR as the augmentation function. There is meaningful variance across classes depending on the task. For sentiment, there are far more new corrections than new mistakes for the positive and neutral classes, but there are slightly more mistakes for the negative sentiment. LLM-TTA only benefits non-toxic examples in the toxicity task. Fewer predictions change overall for news task classes than sentiment and toxicity.

New mistakes dampen the performance gains introduced by TTA. Even classes where TTA overall improves performance can have many new mistakes. Mitigating the trend of two steps forward and one step back is a promising direction for further improving TTA's effectiveness.

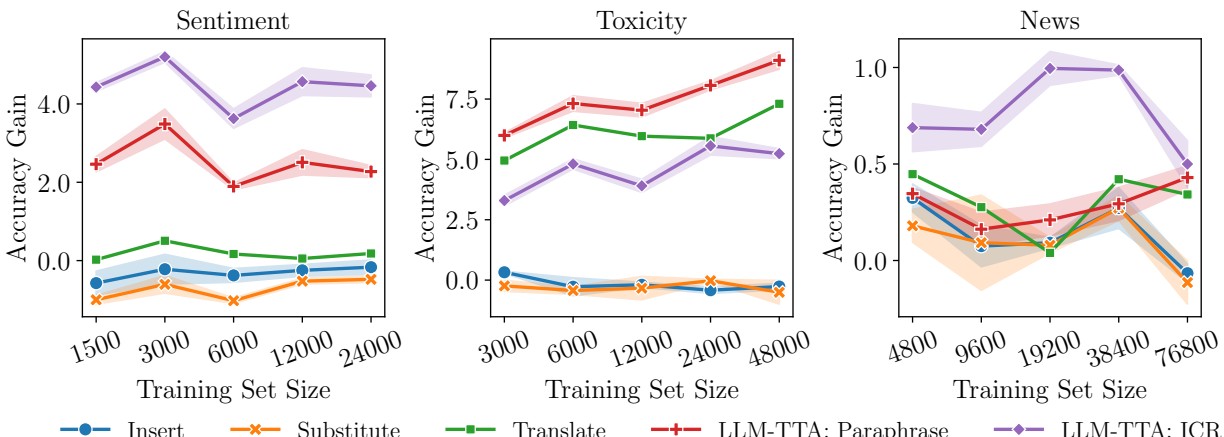

Figure 4: **TTA Effectiveness Across Data Scales.** This figure shows the absolute improvements in OOD accuracy averaged across shifts and experiment runs with standard deviations. We train five BERT models on 5%, 10%, 20%, 40%, and 80% of the ID training set. We find that LLM-TTA improves robustness across data scales. These results suggest that LLM-TTA can still be helpful for practitioners operating in data-scarce regimes.

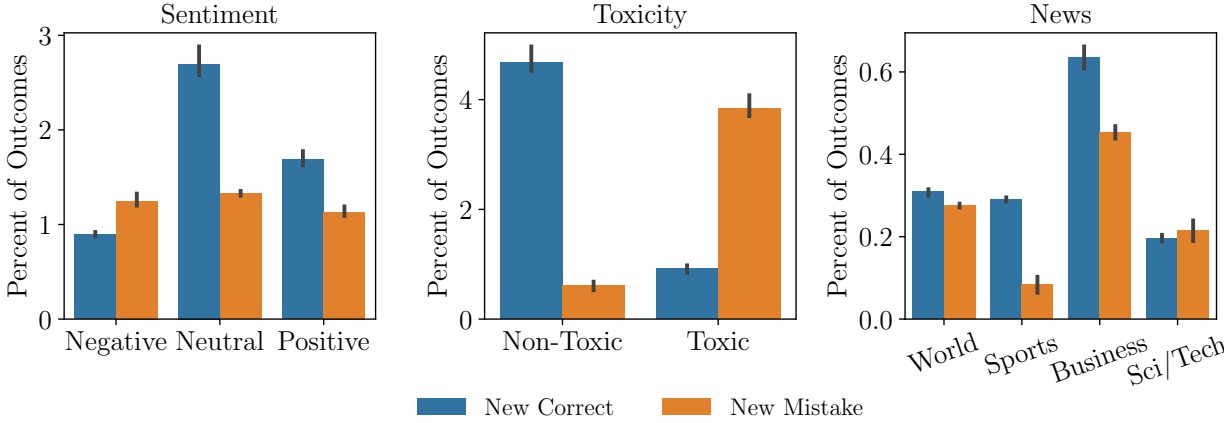

Figure 5: **Changed Predictions Across Classes.** Results are from BERT with ICR as the TTA augmentation function across all OOD inputs. Variance across classes indicates that TTA affects some classes more than others. TTA can hurt the performance of some classes while improving overall performance.

### 5.7 The Optimal Number of Augmentations Varies by Task

The number of augmentations generated per test input is an important hyperparameter in TTA. Determining an optimal augmentation count that balances performance improvements and efficiency is critical when augmentation is expensive, as with LLM-TTA. In this experiment, we study BERT's OOD robustness averaged across OOD shifts using the test input with varying numbers of augmentations.

Figure 6 reports performance across augmentation counts. LLM-TTA's performance largely plateaus after two augmentations, whereas word-level augmentation functions can benefit from larger augmentation batches. These results demonstrate that practitioners may be able to improve efficiency by using fewer augmentations without compromising performance.

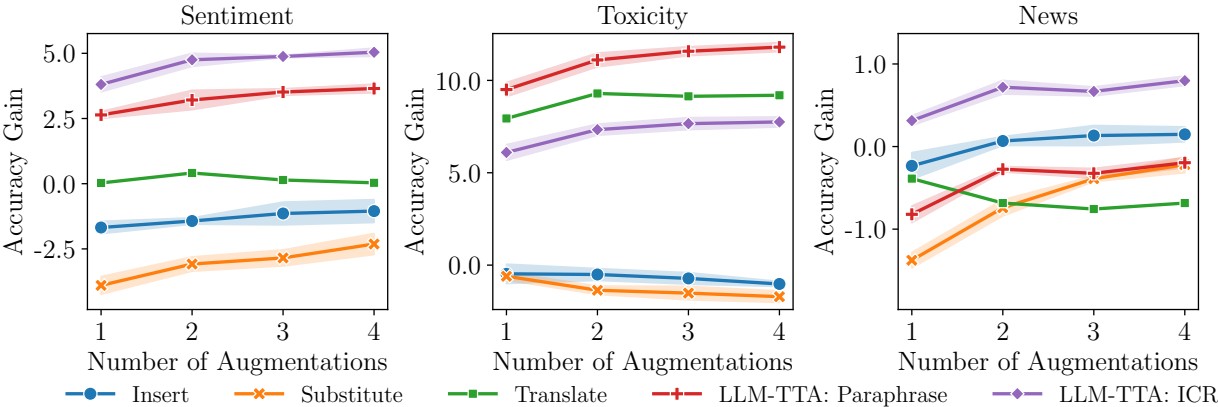

Figure 6: **TTA Effectiveness Across Augmentation Count.** We report the improvements in OOD accuracy averaged across shifts for each augmentation function and the number of augmentations used per inference. LLM-TTA's performance generally plateaus after two augmentations per inference.

## 6    Limitations

We study how well LLM-TTA can improve performance without investigating the specific factors contributing to the datasets being OOD — just that models struggle to generalize to them. Other works have proposed a more nuanced view of OOD robustness (Hendrycks et al., 2020a; Koh et al., 2020; Taori et al., 2020). A common alternative approach to studying OOD robustness is to control for the shifts by modifying specific properties of the training and evaluation sets, such as increasing length (Varis & Bojar, 2021; Ruoss et al., 2023; Zhou et al., 2024), introducing perturbations (Hendrycks & Dietterich, 2019), and evaluating on examples harder than those seen in training (Hase et al., 2024). In contrast, our evaluation sets are selected based on low cosine similarity with ID centroids or synthetically created writing style shifts. These criteria provide an opportunity to study generalization more broadly but do not rule out the possibility that the evaluation data contains challenging ID samples. Future work could further isolate and measure the properties that make datasets OOD to better understand which types of shifts LLM-TTA can most effectively improve.

LLM-TTA outperforms TTA with conventional augmentation functions across tasks for most task models we study. However, these gains come at the cost of increasing the computation and latency required for generating augmentations. This tradeoff diminishes the LLM-TTA's utility in low-compute settings. LLM-TTA's performance improvements are generally modest and insufficient for undoing the performance regression caused by OOD shifts, even when leveraging a capable LLM for augmentation. Furthermore, it may be practical in some settings to use the LLM directly for the task instead of augmentation if it is superior at the task (Section 5.3).

Larger task models tend to benefit less from LLM-TTA. It remains unclear whether the size of the pretraining corpus, model architecture, or another factor influences how well a task model will respond to LLM-TTA. This trend suggests that LLM-TTA is unlikely to be effective for practitioners using LLMs as their task models. We study the effect that parameter count has in isolation in Appendix A.2. An avenue for future work is to better understand what factors in a task model's training or architecture influence TTA's effectiveness.

We study ESA to avoid augmenting examples that are unlikely to benefit from TTA. This selective augmentation reduces the rate of expensive LLM augmentation while still improving robustness. While entropy is a predictive heuristic of whether the task model will change its prediction, accuracy with selective augmentation underperforms the default approach of augmenting every test input, thus producing a tradeoff between classification performance and efficiency. More work is necessary to better classify which examples will likely benefit from augmentation and explore alternative metrics to entropy.

Our study uses a realistic selection of models yet has limitations. We leverage extensively pretrained BERT, T5, and Falcon models. There is a risk that our evaluation datasets were present in these model's pretraining corpora. It is unclear whether if a model was pretrained or not improves or regresses LLM-TTA's performance. There is some evidence that diverse pretraining can improve a model's natural robustness (Hendrycks et al., 2020b). A better understanding of the effect that pretraining has on TTA is an opportunity for future study.

This work studies TTA solely for short-form text classification. While LLM-TTA outperforms baselines across multiple differing domains, it is unclear how well our results generalize to other NLP tasks. For instance, tasks such as extractive QA (Rajpurkar et al., 2016), which require the original structure of the text to remain unchanged, are unlikely to benefit from augmentation. Studying LLM-TTA in other settings is a promising direction for future work.

## 7 Conclusion

This work studied whether test-time augmentation with LLMs can improve the robustness of task-specific models. We observed positive results — LLM-TTA can improve the robustness and, in some cases, ID performance of task models. LLM-TTA is useful in low and high-resource settings across multiple tasks. These gains are realized without assumptions with respect to the task model's architecture or the ability to train the model. We can use ID entropy statistics to reduce the number of LLM augmentations required via selective augmentation. These results demonstrate that LLM-TTA can be a simple method for improving the robustness of task-specific models, an important problem in applying machine learning to real-world settings. While LLM-TTA does improve robustness, more work is necessary to make task models fully robust to distribution shifts.

## 8 Acknowledgement

We are grateful to EleutherAI for permitting access to their compute resources for initial experiments. The welcome and open research community on the EleutherAI Discord was especially helpful for the literature review, debugging PyTorch issues, and information necessary to conduct the parameter count ablation experiment (Appendix A.2). In particular, we would like to thank Stella Biderman, Nora Belrose, and Hailey Schoelkopf.

We are grateful to the University of Virginia Research Computing team for providing access to excellent high-performance computing resources.

Lydia O'Brien provided copy editing and feedback on figure design and engaged in extensive discussions that shaped the direction of this project.

M. Ghassemi's work is supported in part by Quanta Computing and the Gordon and Betty Moore Foundation. The research of J. Mendez-Mendez is funded by an MIT-IBM Distinguished Postdoctoral Fellowship.

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

# A Appendix

## A.1 Detailed Main Results

| Model | Augmentation | Boss Sentiment | | | | Boss Toxicity | | | | AG News → Tweets | |
|---|---|---|---|---|---|---|---|---|---|---|---|
| | | AZ* | SST | SE | DS | CC* | AC | TG | IH | News* | Tweets |
| BERT | Paraphrase | 90.54% | 72.10% | 47.95% | 46.48% | 90.58% | 63.29% | 66.14% | 64.96% | 93.72% | 89.03% |
| | ICR | 90.85% | 73.60% | 49.02% | 48.56% | 91.45% | 51.45% | 66.14% | 65.64% | 93.53% | 89.66% |
| T5 | Paraphrase | 89.90% | 77.06% | 50.86% | 50.30% | 91.32% | 71.62% | 66.35% | 64.42% | 92.05% | 89.84% |
| | ICR | 90.78% | 75.75% | 51.11% | 52.27% | 91.87% | 59.90% | 66.24% | 64.93% | 91.99% | 90.23% |
| Falcon | Paraphrase | 90.35% | 53.93% | 40.45% | 41.57% | 18.43% | 79.95% | 52.97% | 43.53% | 28.78% | 27.65% |
| | ICR | 90.35% | 51.03% | 39.74% | 40.42% | 19.00% | 77.90% | 52.44% | 44.45% | 29.47% | 27.60% |

Table 5: **Seed = 3.** LLM-TTA Accuracy across augmentation functions, datasets, and models. "*" indicates the ID split the task models are optimized for.

| Model | Augmentation | Boss Sentiment | | | | Boss Toxicity | | | | AG News → Tweets | |
|---|---|---|---|---|---|---|---|---|---|---|---|
| | | AZ* | SST | SE | DS | CC* | AC | TG | IH | News* | Tweets |
| BERT | Paraphrase | 90.54% | 71.54% | 47.98% | 45.86% | 90.58% | 62.44% | 66.03% | 65.10% | 93.72% | 88.95% |
| | ICR | 90.85% | 73.03% | 49.23% | 48.80% | 91.45% | 51.57% | 65.92% | 65.74% | 93.53% | 89.73% |
| T5 | Paraphrase | 89.90% | 75.94% | 50.84% | 49.75% | 91.32% | 71.98% | 66.56% | 64.57% | 92.05% | 89.90% |
| | ICR | 90.78% | 75.19% | 51.16% | 52.22% | 91.87% | 60.63% | 66.67% | 65.14% | 91.99% | 90.25% |
| Falcon | Paraphrase | 90.35% | 53.00% | 40.64% | 41.48% | 18.43% | 80.31% | 52.76% | 43.62% | 28.78% | 27.56% |
| | ICR | 90.35% | 51.22% | 39.70% | 40.56% | 19.00% | 77.78% | 52.12% | 44.31% | 29.47% | 27.83% |

Table 6: **Seed = 17.**

| Model | Augmentation | Boss Sentiment | | | | Boss Toxicity | | | | AG News → Tweets | |
|---|---|---|---|---|---|---|---|---|---|---|---|
| | | AZ* | SST | SE | DS | CC* | AC | TG | IH | News* | Tweets |
| BERT | Paraphrase | 90.54% | 71.63% | 47.94% | 46.00% | 90.58% | 64.13% | 65.92% | 65.03% | 93.72% | 88.95% |
| | ICR | 90.85% | 73.50% | 49.21% | 48.96% | 91.45% | 51.69% | 64.97% | 65.75% | 93.53% | 89.81% |
| T5 | Paraphrase | 89.90% | 76.59% | 50.51% | 49.61% | 91.32% | 70.17% | 66.67% | 64.44% | 92.05% | 90.07% |
| | ICR | 90.78% | 76.12% | 51.29% | 52.45% | 91.87% | 59.54% | 66.24% | 64.91% | 91.99% | 90.53% |
| Falcon | Paraphrase | 90.35% | 53.09% | 40.71% | 41.11% | 18.43% | 80.43% | 52.97% | 43.70% | 28.78% | 27.26% |
| | ICR | 90.35% | 50.94% | 39.73% | 41.09% | 19.00% | 77.17% | 52.44% | 44.26% | 29.47% | 27.83% |

Table 7: **Seed = 46.**

## A.2 Isolating Parameter Count's Influence on TTA Performance

Table 1 shows that BERT, the smallest model evaluated in terms of parameter count, benefits the most from TTA. That T5 and Falcon did not benefit as consistently raises the question of whether TTA becomes less effective with model scale. This effect can only be imperfectly studied with BERT, T5, and Falcon since there are confounding factors such as architecture and training datasets beyond parameter count.

The Pythia model suite (Biderman et al., 2023) is a suite of decoder-only transformer languages models trained on The Pile (Gao et al., 2020). Each model in the suite is identical in architecture, training data, and data ordering, with parameter count as the salient difference. Although Pythia is not state-of-the-art, the reduced confounders help isolate the influence that model size has on TTA performance. We evaluate TTA performance on 2.8b, 6.9b, and 12b Pythia models trained with duplicated examples.

Table 9 reports performance across model sizes. We do not observe a clear relationship between parameter count and the degree to which TTA improves performance. This result is observed for traditional and LLM-based TTA methods. Pythia (with duplicated training example) was trained on 300 billion tokens, while

| Model | Augmentation | Boss Sentiment | | | | Boss Toxicity | | | | AG News → Tweets | |
|---|---|---|---|---|---|---|---|---|---|---|---|
| | | AZ* | SST | SE | DS | CC* | AC | TG | IH | News* | Tweets |
| BERT | Paraphrase | 90.54% | 71.54% | 48.11% | 45.74% | 90.58% | 62.08% | 66.03% | 64.93% | 93.72% | 88.94% |
| | ICR | 90.85% | 73.78% | 49.24% | 49.05% | 91.45% | 52.42% | 65.82% | 65.65% | 93.53% | 89.70% |
| T5 | Paraphrase | 89.90% | 76.12% | 50.53% | 49.98% | 91.32% | 69.08% | 66.24% | 64.35% | 92.05% | 90.12% |
| | ICR | 90.78% | 75.66% | 50.97% | 52.78% | 91.87% | 59.78% | 67.20% | 64.81% | 91.99% | 90.20% |
| Falcon | Paraphrase | 90.35% | 53.75% | 40.72% | 41.37% | 18.43% | 80.07% | 52.87% | 43.64% | 28.78% | 27.55% |
| | ICR | 90.35% | 50.94% | 39.52% | 40.58% | 19.00% | 78.14% | 52.02% | 44.23% | 29.47% | 27.99% |

Table 8: **Seed = 58.**

| Augmentation | SST-5 | | | ToxiGen | | | AG Tweets | | |
|---|---|---|---|---|---|---|---|---|---|
| | 2.8b | 6.9b | 12b | 2.8b | 6.9b | 12b | 2.8b | 6.9b | 12b |
| None | 39.27% | 51.96% | 46.18% | 57.63% | 57.63% | 58.90% | 25.16% | 25.05% | 25.14% |
| Insert | 39.37% | 51.12% | 45.99% | 57.61% | 57.63% | 59.00% | 25.04% | 24.99% | 25.03% |
| Substitute | 39.46% | 49.81% | 46.55% | 57.63% | 57.63% | 57.42% | 25.03% | 24.99% | 24.99% |
| Translate | 40.86% | 51.49% | 45.52% | 57.63% | 57.63% | 58.79% | 25.17% | 25.04% | 25.04% |
| Paraphrase | 41.20% | 51.12% | 43.82% | 57.55% | 57.55% | 58.40% | 25.30% | 25.11% | 25.11% |
| ICR | 40.86% | 53.08% | 45.05% | 56.63% | 57.63% | 58.16% | 25.18% | 25.12% | 25.18% |

Table 9: **Parameter Count Ablation** TTA accuracy across Pythia model scale is reported. The degree to which TTA improves performance over the baseline is not strongly correlated with parameter count. These findings suggest that other factors, such as the training distribution and model architecture, are more likely to influence TTA's performance.

BERT was trained on 3.3 billion tokens (Clark et al., 2019). We conjecture that the pertaining objective and corpus size may play a larger role than the learnable parameter count in isolation.

### A.3 Analyzing In-Context Rewriting Augmentations

In-Context Rewriting (ICR) prompts the LLM to rewrite OOD examples such that they are more like a set of ID examples provided in the prompt. The intuition behind this technique is that the strong ID performance can be generalized to OOD examples, which are rewritten such they are ID while preserving semantics. Does ICR successfully rewrite OOD examples such that they're ID? We study this question by evaluating the embeddings of the ID evaluation set for Boss Sentiment and the SST-5 OOD shift. The mean representation of the four augmentations for each OOD input is used. We use the RoBERTa[4] model introduced in Gao et al. (2021) for embeddings.

ICR's augmentations generally do not bridge the gap between distributions. The cosine similarity between the ID evaluation set's centroids and the SST-5 original examples centroid is **0.3285**. The similarity between the augmentation's centroid and the ID eval set is **0.3445** — only slightly more similar. Figure 7 demonstrates that most augmentations remain OOD.

These results suggest that prompting may be insufficient for entirely bridging distribution gaps. However, the fact that ICR consistently outperforms 0-shot paraphrasing suggests that ID examples in the prompt still positively influence performance. Additional work is needed to bridge domain gaps fully.

### A.4 Selecting Entropy Thresholds: Expanded

Section 5.4 describes entropy-based selective augmentation. Selecting the entropy threshold involves deciding which distribution to sample from and the tradeoff between augmentation rate and performance. We use the ID evaluation set to find an optimal threshold. We selected the entropy based on a generalization of the F-score metric. We treated accuracy as precision and the augmentation rate as recall.

---

[4]https://huggingface.co/princeton-nlp/sup-simcse-roberta-large

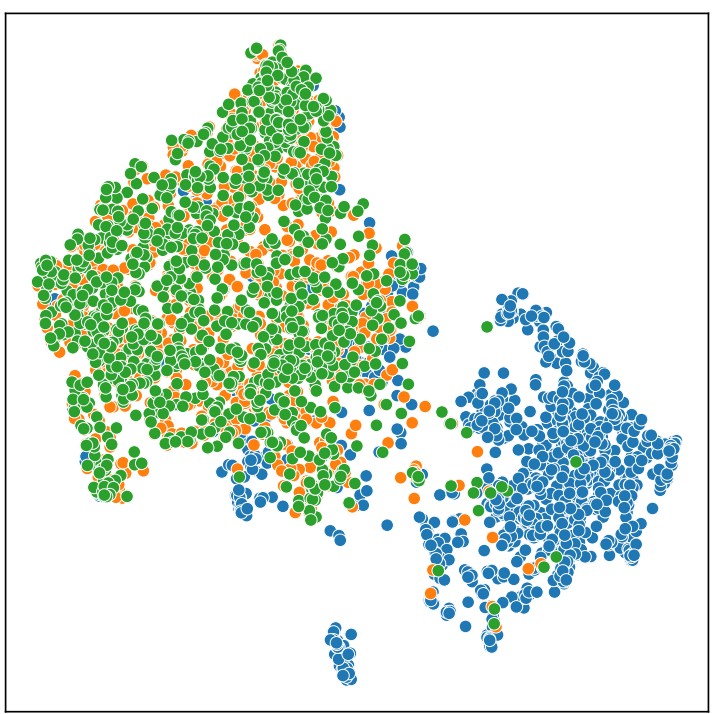

•  In-Distribution          •  Out-of-Distribution          •  Out-of-Distribution Augmented

Figure 7: **2-D UMAP Representations of Text Embeddings.** Embeddings for the Boss Sentiment ID evaluation set (Amazon reviews), the OOD (SST-5) test examples, and the mean embeddings for each test inputs augmentation batch are plotted. Suppose ICR was successfully rewriting OOD examples to be ID. In that case, we would expect to see far more augmentations in the same region of embedding space as the ID evaluation. The fact that augmentations remain largely in the same space as the OOD inputs they are sourced from suggests that ICR is not bridging the domain gap, even in cases where ICR meaningfully improves performance.

Let $rate_{aug}$ be the augmentation rate (the percent of examples augmented) and $a$ by the accuracy of the given distribution.

$$rate_{aug} = (1 + (\frac{1}{500})^2) \cdot \frac{\text{acc} \cdot (1 \text{ - } rate_{aug})}{((\frac{1}{500})^2 \cdot \text{acc}) + (1 \text{ - } rate_{aug})} \tag{2}$$

### A.5  Task Model Training Details

We opted to train our own task-specific models for the experiments to have maximal control over the training process. We use an uncased BERT base model available through the HuggingFace Transformers library (Wolf et al., 2019). We trained a separate model on each separate in-distribution dataset for a total of four models. Following the suggested baseline in Mosbach et al. (2020), we used a batch size of 32 examples, weight decay of 0.01, and a linear learning rate schedule peaking at 2e-5. We took the checkpoint that scored the highest average class F1 on the ID test set. T5-Large followed the same training procedure and identical hyperparameters.

### A.6  Dataset Statistics

| Amazon (ID) | | | | SST-5 | | | | Sem Eval | | | | Dynasent | | | |
|---|---|---|---|---|---|---|---|---|---|---|---|---|---|---|---|
| Neg | Neutral | Pos | Total | Neg | Neutral | Pos | Total | Neg | Neutral | Pos | Total | Neg | Neutral | Pos | Total |
| 2181 | 32862 | 3861 | 38904 | 282 | 400 | 390 | 1072 | 3229 | 7059 | 10336 | 20624 | 1440 | 1440 | 1440 | 4320 |

Table 10: Sentiment Task Dataset Statistics

| Civil Comments (ID) | | | Adv. Civil | | | ToxiGen | | | Implicit Hate | | |
|---|---|---|---|---|---|---|---|---|---|---|---|
| Benign | Toxic | Total | Benign | Toxic | Total | Benign | Toxic | Total | Benign | Toxic | Total |
| 89543 | 7777 | 97320 | 152 | 672 | 824 | 544 | 400 | 944 | 13291 | 8189 | 21480 |

Table 11: Toxicity Task Dataset Statistics

| News (ID) | | | | | Tweets | | | | |
|---|---|---|---|---|---|---|---|---|---|
| Worlds | Sports | Business | Sci/Tech | Total | Worlds | Sports | Business | Sci/Tech | Total |
| 1900 | 1900 | 1900 | 1900 | 7600 | 1900 | 1900 | 1900 | 1900 | 7600 |

Table 12: News Task Dataset Statistics

## A.7   AG News Tweets Dataset

**Motivation**   AG News is a four-way topic classification task introduced in Zhang et al. (2015). In this setup, a task model must classify whether a given news article is about world events (***World***), sports and athletics (***Sports***), business and economics (***Business***), and scientific developments (***Sci/Tech***). The test set on HuggingFace (`huggingface.co/datasets/ag_news`) is composed of 7,600 examples equally balanced across the four classes.

News topic classification presents a promising opportunity for largely isolating the effect of writing style shifts. Existing deep learning methods also perform well on this dataset with accuracy reaching higher than 90% (`paperswithcode.com/sota/text-classification-on-ag-news`).

Another motivation for this particular task is the common risk of data augmentation inadvertently flipping the label/semantics of the text Bayer et al. (2021). Unlike other tasks such as sentiment classification or subtle hate speech, the topic of a news article is unlikely to change during augmentation, thus preserving the original label.

**Creation**   We used GPT-3.5 Turbo Brown et al. (2020) (6/7/23 version) for style transfer. We did an initial pass through all 7,600 examples using a conservative "V1" prompt and greedy decoding. Calls were made using the OpenAI Python SDK with top_p and temperature set to zero. The data was then lightly preprocessed to reduce the number of examples that began with **BREAKING NEWS** flanked my emojis. The V1 prompt is *Write the following news summary in the style of a Twitter/social media. Maintain all the relevant information and sentiment*", and the V2 prompt is "*Write the following news summary in the style of a Twitter/- social media. Maintain all the relevant information and senti- ment. Add some flare with humor, anger, or sarcasm*".

512 of the initial model responses did not result in satisfactory generations. These were typical cases where the generated text was almost indiscernible from the original text or the generation was entirely emojis. We called GPT-3.5 Turbo again with an updated prompt and hyperparameters (temperature=0.7, top_p=0.9, frequency_penalty=0.5, presence_penalty=0.5) for these examples. Whereas all the first-pass generations did not have any instructions to the model as to the sentiment/mood of the hypothetical post author, we purposefully instructed the model to "*Add some flare with humor, anger, or sarcasm.*" in the generation.

It's important to note that we did not enforce Twitter's character limit. These sequences should be considered as more broadly inspired by social media posts rather than following the exact specifications of Twitter

posts. We also did not manually review every sequence in the dataset to confirm that the original label was preserved. GPT 3.5 Turbo also hallucinates facts, such as adding the hashtag #Olympics2021 even though the original dataset was created in 2015.

## A.8 Examples

| Version 1 Prompt Examples | | |
|---|---|---|
| Label | Original | Generated |
| World | Venezuela Prepares for Chavez Recall Vote Supporters and rivals warn of possible fraud; government says Chavez's defeat could produce turmoil in world oil market. | #Breaking: Venezuela readies for #ChavezRecallVote amidst cries of #fraud from both sides. Government cautions defeat could shake up #oilmarket. |
| Sports | Natalie Coughlin Wins 100M Backstroke (AP) AP - American Natalie Coughlin won Olympic gold in the 100-meter backstroke Monday night. Coughlin, the only woman ever to swim under 1 minute in the event, finished first in 1 minute, 0.37 seconds. Kirsty Coventry of Zimbabwe, who swims at Auburn University in Alabama, earned the silver in 1:00.50. Laure Manaudou of France took bronze in 1:00.88. | Natalie Coughlin makes history as she wins gold in the 100M backstroke at the #Olympics2021. She finishes in 1 minute 0.37 seconds - her incredible performance leading her to become the only woman to swim under 1 minute in the event. Congrats, Natalie! #TeamUSA |
| Business | Dollar Near Recent Lows, Awaits ZEW/CPI LONDON (Reuters) - The dollar held steady near this week's four-week low against the euro on Tuesday with investors awaiting a German investor confidence survey and U.S. consumer inflation numbers to shed light on the direction. | Dollar stays near recent lows against the euro Investors await German survey & US consumer inflation data Info to give direction to the currency #finance #currencyexchange #investing |
| Sci/Tech | Intel Delays Launch of Projection TV Chip In another product postponement, semiconductor giant Intel Corp. said it won't be offering a chip for projection TVs by the end of 2004 as it had announced earlier this year. | #TechNews: Intel announces delay in launch of projection TV chip. Chip won't be available by end of 2004 as previously announced. #Intel #Delay #TVChip |

| Version 2 Prompt Examples | | |
|---|---|---|
| Label | Original | Generated |
| World | Venezuela vote keeps Chavez as president CARACAS – Venezuelans voted resoundingly to keep firebrand populist Hugo Chavez as their president in a victory that drew noisy reactions yesterday from both sides in the streets. International observers certified the results as clean and accurate. | Hugo Chavez remains as the President of Venezuela! The people have spoken, and they want their firebrand populist leader to continue leading the country. *Cue street celebrations* Oh, and just a little side note, international observers gave the results a for being clean and accurate. #ChavezForever #VivaLaVenezuela |
| Sports | St. Louis Cardinals News Right-hander Matt Morris threw seven solid innings, but the Cardinals needed a bases-loaded walk to second baseman Tony Womack and a grand slam from new right fielder Larry Walker to key a six-run eighth inning for a ... | Matt Morris doing the heavy lifting for the @Cardinals, but it was Tony Womack's patience and Larry Walker's big swing that sealed the deal with a SIX run eighth inning! #OpeningDay #RedbirdsRising |
| Business | Yahoo! Ups Ante for Small Businesses Web giant Yahoo! is gambling that price cuts on its domain name registration and Web hosting products will make it more competitive with discounters in the space – which means that small businesses looking to move online get a sweeter deal through ... | NEWS ALERT Yahoo! is stepping up its game for small businesses! The web giant is slashing prices on domain name registration and web hosting products to keep up with competitors. Now, small biz owners can get a sweeter deal through Yahoo!. #Yahoo #SmallBusiness #WebHosting |
| Sci/Tech | Most US homes have broadband connections While the total number of home internet users has reached a plateau in the US, those who do use the internet are adopting broadband at a rapid pace, according to Marc Ryan, senior director of analysis at the audience measurement company. | Good news, America! Most homes now have broadband connections! While the total number of internet users has plateaued, those online are adopting broadband at lightning speed. According to Marc Ryan at @audmeasurement, it's time to kick dial-up to the curb! #InternetUpgrade #NoMoreDialUp |

## A.9 LLM Inference Parameters

**TTA: OOD Generations.** For LLM-TTA and back-translation, we generate four augmentations for each test input using temperature-based decoding with a temperature of 0.3.

---

**Sentiment Prompt**

```
Task: The following text has either a negative (0), neutral (2), or positive (1). Return the sentiment label for the text.

"worked OK, Kind of pricey" - Label=2

"Einstein defined idiocy as doing the same thing over and over and yet expecting a different outcome. A definition I was reminded of after the latest
Must-be-OnLine-to-play disaster. If there ever were an attempt to prove that there is, indeed, bad publicity, the release of SimCity 2013 sure was it.
Mega-publishers are repeatedly trying to turn the beautiful artform of gaming into a utility, where "gaming content" will be streaming to your TV or PC or phone
- and you will be charged by the second for it. Monopolistic issues aside for the moment, are they even remotely ready for" - Label=0

"Well researched book. I believed most of it although you could nod off during some of the details." - Label=2

"That being said, it's possibilities dropped WAY down and i'm going to return it. The plug-in microphone seems like a situation just asking for a "broken" piece.
The sound quality is average, at best, and i still think my After Glow headset is way better.  If you're looking for a complete 100% wireless
experience, i think Turtle Beach may be the way you need to go." - Label=2

"Good movie." - Label=2

"Very nice and clean feeling after use1" - Label=1

"Great buy" - Label=1

"The material is good, but the book was maybe not originally written in English? I found grammar, spelling, and punctuation mistakes aplenty in my Kindle version.
If one is patient while reading the information can be gleaned. The subject is very interesting and if you have trouble taking man made anti-biotics you will
find the book to be very helpful. I have personal experience in using many of the herbs listed and can speak to their high efficacy." - Label=2

"They go not look like the picture, I know jewelry, not very good quality. I may return them" - Label=0

"NOT 30" deep. Maybe 24"? False advertising!" - Label=0

"Good product, great price. I've bought about 20 of them now...all well made." - Label=1

"Extremely strong and durable power cable. Much better quality than the included apple lightning cables that I've used.  +Works well with powerbanks and 2.1a
chargers. +My cable doesn't get garbled up with the Nylon braiding +It charges my Ipad air 2 at full speed.  Great product." - Label=1

"Was a little disappointed in how easily the vegetables fall out of the holder so haven't really used it very much" - Label=2

"We love Pete the Cat in our house! We were super excited to see what he would be up to in this new adventure. This time he loves his shirt with the four groovy
buttons. One by one they all fall off! But Pete doesn't let that bother him, he just keeps singing his little song, with a little twist!" - Label=1

"I have a shower with white-to very light gray slab granite, and a tub deck made of the same. This granite absorbs moisture and darkens. I used this sealer,
and am not impressed. Let me preface my comments by saying that I squeegee AND wipe down all the granite 100% of the time. Right off the bat, the tub deck had a
couple of places where the sealer didn't seem to do much of anything. After 3 months, more spots are developing that show this moisture-induced darkening.
The sealer is clearly losing its effectiveness, after only 3 months. Very disappointing," - Label=0

"OtterBox makes a great product." - Label=1

What is the label for the following text? You must decide which label the text is."

"<test_input>" - Label=
```

**TTA: ID Generations.** LLM-TTA uses beam search decoding four return sequences, four beam groups, four beams, and a diversity penalty of 0.5. Back-translation uses the HuggingFace generation pipeline API. Text is translated from English to German (facebook/wmt19-en-de) and then back into English (facebook/wmt19-de-en). The generation parameters are four return sequences, temperature of 0.7, four beams, four beam groups, top-P of 0.95, top_k of 0, repetition penalty of 10.0, diversity penalty of 1.0, and no repeating n-gram size of 2.

**LLM Classifier Inference.** We use the following prompt template for LLM inference. Each prompt contains 16 in-distribution training examples selected and ordered randomly within the prompt with a random seed of 42. Examples in the prompt are balanced across classes. Greedy decoding is used with a max of 10 new tokens. Verbalalizers are used to map generated tokens to labels.

## Toxicity Prompt

Task: Given a piece of text, your task is to classify whether it contains toxic or not. If the text is not toxic, return 0. If it is toxic, return 1.

"The economic powerhouse for Robeson County, tobacco, was sent off shore by the law suites executed by the Bill Clinton DOJ. Now, the tobacco is still grown, just in Brazil, not in my home town of Fairmont. Billions of $'s went to the lawyer buddies of Clinton, but this has left Eastern North Carolina destitute. We used to make 50k/yr. off tobacco on our farm during the '80's. Want to know how that land is used today? It is a set-aside for natural dove (yes, birds) habitat. Just pays the taxes." - Label=0

"I don't see our country retreating into isolationism. The world won't allow it. I expect Trump will be tested by our adversaries early into his presidency. How he reacts is anybodies guess. but i don't see a whole lot of back up happening. My guess is the world will either come to respect him or fear him. In the world of geo-politics both appear to have equal value." - Label=0

"I don't need advice from a right wing Republican about whether I should socialize with right wing Republicans. Why doesn't he address himself to whether the Attorney General of the United States should be giving secret speeches to anti-gay hate groups? Presumably some on the right might actually be interested in his opinion about that. Trump isn't just a conservative, he's a criminal and a traitor. Anyone who supports him is either a fool or a scoundrel. If you support Trump, I don't need to find out which of those types you are before I tell you to take a" - Label=1

"Well Mr. Eddie, it sucks to be you. I have worked in the office building oil industry for decades and only on the rarest of occasions have encountered the poison type of environment you describe. Sure, there are loads of personalities but the back stabbing and blatant ladder climbing is the exception rather than the rule. Sure, the stakes are higher; your capabilities are being evaluated. You are hired for role; how well do you fill it.? Maybe there is is little teamwork in journalism; that would explain your experience. A team quickly becomes dysfunctional if everybody is out for" - Label=1

"You left something out. What does the russian diplomat offer in return for the repeal of sanctions?" - Label=0

"They'll say why did it take them so long to tackle greenhouse gas emissions when it was obvious much earlier that something serious had to be done." - Label=0

"I might stop to give somebody a boost but not to change a wheel. If you can't do that by yourself you shouldn't be driving." - Label=0

"Yes..there is actual talent...then there is talent at backstabbing, sucking up, and cutting someone else's throat... The merit should lie with the task at hand...but unfortunately that is rare these days." - Label=1

"Does it take a certain special political gene to already be present in a person that makes them slimy, opportunistic and giant hypocrites? Or is that a learned trait once they get voted into the pig trough and see everyone else vacuuming up tax money and selling themselves to the highest bidder?  I've always wondered." - Label=1

"It's okay. It seems Canadians are content with minimum and mediocre wages, simple and boring life styles. Let the rest of the world reap the dragon, to become rich and prosperous. We are not known for having global perspectives, ambitious economic plans. Our gut fear of an ideology different from our own makes us an introverted bunch. After all, life would be all good if all of us can live just slightly above the poverty line. We don't need your Chinese money." - Label=1

"As an atheist, I find this letter as a good reminder not to lump all Christians together in terms of their stance on such matters. Common ground can be found if we're willing to search for it." - Label=0

"He said legal fees - not government legal fees. That includes everyone. Pay attention." - Label=0

"If anyone in Trump's administration had said that the Comey firing was "wrong", they too would be fired by Trump. That's just the way he operates. He's a selfish, arrogant, clueless, and completely unqualified politician in every aspect. It's a very sad era in the history of our country, and the sooner Donald Trump is removed from office, the sooner we can get our country back on track. Even the majority of Trump voters know that they made a huge mistake in giving him their vote." - Label=0

"You are an idiot. You think Facebook elected Trump?" - Label=1

"She calls Trumpster a blow hard, Hillary a liar and people who vote by what the polls say idiots." - Label=1

"As do Trump's trade czar, the head of the British Commonwealth, a hypocritical pop star and our honoured conservative ex PM.  This isn't about Liberal or Conservative. It's about the privileged taking advantage of loopholes because they can afford the legal advice the hoi polloi cannot." - Label=1

What is the label for the following text? You must decide which label the text is."

"<test_input>" - Label=

## News Topic Prompt

Task: Given the news text, classify whether the topic of the article is of the following. Return the integer ID for each category.

World = 0, Sports=1, Business = 2, Sci/Tech=3

"Security tight for Afghan inauguration Afghan and international forces laid a security ring around Kabul on Monday in preparation for the inauguration of Hamid Karzai as Afghanistan #39;s first popularly elected" - Label=0

"Ukraine Opposition Flexes Muscles After Victory  KIEV (Reuters) - Opposition leader Viktor Yushchenko, fresh  from securing a re-run of Ukraine's disputed election, tore up  a deal on Saturday to reduce the president's powers and told  his supporters
to keep up pressure on the streets." - Label=0

"A Hot Stock at 7 Times Earnings Seaboard, a small agribusiness and transportation company, looks promising." - Label=2

"Giants Hold On to Warner The Giants on Thursday tried to fan away every possible wisp of a quarterback crisis.
Even rookie Eli Manning tried to do his part." - Label=1

"Clemens considers retirement, but no decision yet  quot;I #39;m very close to retirement, quot; Clemens told reporters Friday.  quot;Obviously,
I haven #39;t made the decision for the upcoming season yet." - Label=1

"Illinois Governor wants to Ban Violent Video games Yesterday Illinois Gov. Rod Blagojevich proposed a state law that would ban the selling and rental
of violent and or sexually explicit video games to Children under the age of 18." - Label=3

"Pilots at US Airways Will Vote on Plan to Cut Pay and Benefits The pilots will vote on US Airway's demand for \$300 million in wage and benefit concessions,
their union said Tuesday night." - Label=2

"Microsoft dodges anti-spyware charge accusations According to a story on CNN #39;s Web site, the software giant #39;s recent acquisition of anti-spyware company GIANT
might mean users have to pay in order to benefit from the added security to Microsoft applications." - Label=3

"Medtronic Wins FDA Approval on Insync Sentury ICD  CHICAGO (Reuters) - Medtronic Inc. <A
HREF="http://www.reuters.co.uk/financeQuoteLookup.jhtml?ticker=MDT.N qtype=sym infotype=info qcat=news">MDT.N</A> on
Monday said  U.S. regulators approved its Insync Sentry implantable  defibrillator (ICD) months earlier than it expected." - Label=3

"Marsh general counsel to resign amid probe NEW YORK - The general counsel for broker Marsh  amp; McLennan Cos. will step down next week
as the embattled firm seeks to clean house and reform its business practices, the Wall Street Journal reported on Friday." - Label=2

"Labour, Likud agree on unity government, eight Palestinians killed ... JERUSALEM (AFP) - Prime Minister Ariel Sharon #39;s Likud
party and the opposition Labour agreed to form a national unity government, while at least eight Palestinians
were killed during an Israeli army incursion in southern Gaza." - Label=0

"Panthers' Davis, Morgan Sit Out (AP) AP - Carolina running back Stephen Davis missed his sixth game of the season Sunday because of a knee injury.
Linebacker Dan Morgan, who sustained a concussion last week, also was out for the Panthers' game against Oakland." - Label=1

"Cairo hosting Arafat funeral A military funeral for the Palestinian President, Yasser Arafat, will be held in the Egyptian capital, Cairo this morning.
After the funeral President Arafat #39;s coffin will be flown by helicopter to Ramallah for burial." - Label=0

"How Can We Stop IM Worms? Traditional antivirus technology may be too slow to protect users, researchers say." - Label=3

"GenCorp rejects Steel Partners offer GenCorp (GY.N: Quote, Profile, Research) , an aerospace and real estate company, on Monday said its board has rejected a
\$17 per share offer from US investment fund Steel Partners II, saying the price was  quot;inadequate." - Label=2

"A Matter of Trust (Forbes.com) Forbes.com - There is more to corporate performance than what you see in the earnings reports. Could an investor have anticipated
the trouble at companies like Enron, Adelphia, WorldCom and Tyco by looking more closely at how they were governed and how they kept their books? Their problems,
to be sure, are far more visible in hindsight, but nonetheless each left telltale signs that all was not well. Robust reported earnings growth at both
Enron and WorldCom was not supported by hard cash. The Adelphia board was stacked with company insiders who turned a blind" - Label=2

What is the label for the following text? You must decide which label the text is."

"<test_input>" - Label=

