# OpenReview forum: "Improving Black-box Robustness with In-Context Rewriting"
_TMLR — Accepted by TMLR_

### Review · Reviewer_M6U3 · 2024-04-04

**Summary Of Contributions:**

In this work, the Authors propose to improve test-time augmentation methods by using an LLM to paraphrase the test example in various styles, then aggregating the results across paraphrases to predict the final result. Through several standard benchmarks on base embedding models like BERT, it is demonstrated that the system tends to improve OOD performance, while not incurring a significant penalty in IID performance.

**Audience:**

Yes

**Claims And Evidence:**

Yes

**Requested Changes:**

Firstly, and critically, it is important to obtain confidence measures around these numbers. While the model weights may remain fixed, we should still be able to measure the performance across different sample collections paraphrased by the LLM, and report the variance in that way. This is, in my opinion, critical for practitioners to understand the robustness of the proposed method.

Secondly, while the work's objective of ameliorating OOD generalisation issues is certainly very welcome, I also find that the work focusses only on OOD splits of freeform text tasks. This in my opinion offers a limited view of OOD; many established OOD robustness papers will instead focus on tasks for which controllable distributions at test time can be generated in certain parameters (e.g. problem size). Such papers will tend to focus on reasoning tasks, which are often not hard to synthetically generate (they could be as simple as, e.g. elementary arithmetic tasks). To name a few possible examples:

* Ruoss et al., Randomized Positional Encodings Boost Length Generalization of Transformers
* Zhou et al., Teaching Algorithmic Reasoning via In-context Learning
* Zhou et al., Transformers Can Achieve Length Generalization But Not Robustly
* Veličković et al., The CLRS Algorithmic Reasoning Benchmark

If the authors want to maintain the OOD narrative as they presently have it, I would strongly recommend they evaluate on a reasoning benchmark in the style of the above, and properly contextualise such works in their related work.

**Strengths And Weaknesses:**

I appreciate the simplicity and accessibility of the method. It is well-motivated, combined with a useful set of experiments, and is likely to be a useful addition to the field.

While I am basically ready to recommend acceptance, I also remarked some issues with the reliability of the experiments (namely, and especially, the lack of error bars), and a somewhat reductionist view of OOD generalisation. I will dive into more detail on both of these points in the Requested Changes section.

---

> ### Author Response · Authors · 2024-05-09
> **Author Response**
>
> Thank you for the detailed review. We have provided our response to your concerns below.
>
> **How consistent are our results across runs?**
>
> Thank you for suggesting that we report confidence measures across multiple experiment runs. We have run our OOD experiments multiple times with non-deterministic augmentation functions. Our OOD tables and figures now report the mean performance across runs and the standard deviation as the confidence measure. We observed minimal variance across experiment runs, with a maximum standard deviation of 0.5 percentage points, reinforcing our technique's superiority over the baselines. Our ID multi-run experiments are still ongoing due to compute constraints. We value your suggestion to strengthen the robustness of our analysis.
>
> **Should we evaluate LLM-TTA on reasoning and synthetic benchmarks?**
>
> We agree that extending our analysis to include other tasks, such as reasoning, could be an interesting extension to our existing experiments. However, extending beyond classification is beyond our scope and other areas warrant dedicated studies. Instead, we focus on studying LLM-TTA across domains within text classification, a common problem area in NLP. As also introduced in the general response above, we have addressed this comment by further highlighting our focus on text classification in the abstract and highlighting that our results may not generalize to other problem areas in the Limitations section.
>
> Your suggestion about controlling for the factors leading to the distribution shift is an interesting direction. We control for shifts in writing style in our News topic classification task, as described in Section 4.1. However, the benchmarks we selected for sentiment and toxicity classification do not include controllable synthetic shifts. We address the possibility of confounding factors stemming from the lack of synthetic controllability and by treating OOD robustness as a homogeneous phenomenon in our Limitations section, where we cite Ruoss et al. (2023) and Zhou et al. (2024) as alternative approaches.

---

> ### Comment · Reviewer_M6U3 · 2024-05-09
> **On the meaning of out-of-distribution**
>
> Thank you for your response. I affirm my support for accepting the work.
>
> While the additional discussion of the limitations of the OOD split are certainly welcome, I find it concerning that it is basically reduced to the following observation, provided only in the limitations section after most of the paper is concluded:
>
> > We study how well LLM-TTA can improve performance **without investigating the specific factors** contributing to the datasets being OOD, just that models struggle to generalize to them.
>
> If you do not have a way of quantifying the factors contributing to a data point being outside of a distribution, arguably, you cannot claim with certainty a dataset is OOD.
>
> Your sentence above seems to imply that having weak generalisation performance on an input <=> this input is OOD.
>
> However this sentence is not true in general, because (a) it is possible to poorly generalise on an IID sample, and (b) it is possible to generalise well on an OOD sample. These kinds of effects are, in fact, common in various reasoning tasks.
>
> The reason why I am concerned about this is that the main body of your paper still emphasises the term "OOD" throughout the work. But why do you need the focus on OOD? In my honest opinion, your paper is still valuable if you just say you improve generalisation performance on hard examples.
>
> If you are not intending to explicilty construct datasets for which we can verifiably claim they are OOD, I would highly recommend rewriting the focus of the paper, to avoid using the term so heavily. A shift in "writing style" is not a robustly-described distribution shift.

---

### Review · Reviewer_LfbM · 2024-04-04

**Summary Of Contributions:**

This paper investigates how to improve the out-of-distribution (OOD) robustness by Test-time augmentation (TTA) for NLP tasks. It proposes LLM-TTA, which uses LLM-generated augmentations as TTA’s augmentation function. In particular, it uses two prompt templates including Paraphrasing and In-Context Rewriting (ICR). It also proposes Entropy-Based Selective Augmentation (ESA) for selectively augment test examples that are likely to benefit from TTA while deferring to the model’s original prediction otherwise. The effectiveness of the proposed method and the claims are well supported by experiments.

**Audience:**

Yes

**Claims And Evidence:**

Yes

**Requested Changes:**

NA

**Strengths And Weaknesses:**

**Strengths:**

This paper is well written and I am glad to read it. The description is clear and the claim is well supported by evidence. The idea using in-context rewriting for TTA is new to me, and also make sense, since the LLM models are powerful in language understanding and generation.
The Entropy-Based Selective Augmentation (ESA) for selectively augment test examples is also a good method to save the computational cost for TTA in practice. This paper also well addresses its limitations.


**Weaknesses:**

When the effectiveness of the proposed methods is supported by experiments, I am not confident whether the experiments on classification tasks are enough, especially under the setting that the models are fine-tuned from BERT/T5. Is it better for training from scratch for supporting the claims? Or considering the other generation tasks (even though the evaluation is more complicated)?

---

> ### Author Response · Authors · 2024-05-09
> **Author Response**
>
> Thank you for reviewing our work. We address each of your points below.
>
> **Should we consider evaluating LLM-TTA on generation tasks?**
>
> We address this consideration in the general response, where we clarify the scope of this work as focusing on text classification. Text classification is a common problem area for which we provide a broad evaluation. We agree that studying LLM-TTA for other tasks is interesting, but we believe that evaluating LLM-TTA on other tasks is beyond the scope of this work. To address this feedback, we have clarified this work’s scope more directly in the abstract. We have also added a paragraph to the Limitations section where we highlight that results on text classification may not extend to other problem areas.
>
> **Is training from scratch more appropriate for our setting?**
>
> If we understand your point correctly, we’d like to clarify that all task-specific BERT/T5 models are trained from scratch in all experiments, while the Falcon model and the rewriting LLM are both prompted.

---

> > ### Comment · Reviewer_LfbM · 2024-05-11
> > **comments on the response**
> >
> > Thanks for the feedback of the authors. My concerns on generation tasks are eliminated, given that this paper constraints its contribution for text classification tasks, even though it also downgrades its contribution.
> >
> > In terms the training from scratch, my meaning is that this paper should consider the classification model being training from scratch (weight randomly initialized, not fine-tuned from BERT/T5), since this paper claims “LLM-TTA is agnostic to the task model architecture”. Even though BERT/T5 is widely used classification model,  a simple classification model (randomly initialized) trained over the classification datasets will be a good support to the effectiveness of LLM-TTA, since it removes the affects of other datasets (used for training BERT/T5).

---

### Review · Reviewer_aT3v · 2024-04-24

**Summary Of Contributions:**

This paper proposes a method to improve the model performance on language tasks of both in-distribution and out of distribution scenario of  a black-box model. The proposed method utilizes the scheme of test-time augmentation, and proposes to use a LLM to rewrite the input as a type of augmentation. Additional analysis is made to reduce the portion of test data that needs LLM augmentation. Performance improvement is observed emperically in multiple scenarios.

**Audience:**

Yes

**Claims And Evidence:**

No

**Requested Changes:**

Need a more systematic way to decide when using LLM augmentation is helpful, and when using the LLM directly for the task is better.

**Strengths And Weaknesses:**

## Strength
1. This work is the first to consider LLM augmentation in the test time augmentation scenario
2. Detailed analysis is provided on the impact of augmentation on finegrained performance of different tasks, and the impact of other factors in the process.

## Weakness
1. From the novelty perspective, there are previous work in both using test time augmentation to improve performance, and to use LLM for augment data generation. The paper 's proposal of using LLM for test time augmentation seems to be a straightforward extension on existing methods
2. The effectiveness of the proposed method is in doubt. From the experiment results, using the test time augmentation does not consistently improve model performance, and the use of LLM for augmentation does not consistently outperforms previous augmentation methods. This suggests a missing link in the proposed method, which is identifying whether test time augmentation is helpful for the scenario. Though the paper proposes a entropy-based selection criteria, apperantly it is not effective enough
3. Using LLM is costly, and with the power of LLM in hand, the use of the original black-box model may not be needed any more. More detailed analysis on if using pretrained LLM or LLM + in-context learning would be adequate to outperform the blackbox models used in this paper, especially in the "out of distribution" cases.

---

> ### Author Response · Authors · 2024-05-09
> **Author Response**
>
> Thank you for your time and consideration in drafting this thorough review. We address your listed weaknesses in the order they were presented.
>
> **Does our work make sufficiently novel contributions over existing works?**
>
> We agree that we build upon existing works. Beyond introducing straightforward LLM-based TTA methods, our evaluations are more comprehensive than recent works in that we study a broader set of models, datasets, ane augmentation functions. In particular, our study shines light on the importance (or lack thereof) of the augmentation function’s strength. We believe our findings can help guide future research in this area.
>
> **Is LLM-TTA consistently effective?**
>
> We want to clarify a potential misunderstanding. As shown in Table 1, LLM-TTA outperforms all other methods for BERT and T5 on OOD tasks. Only for Falcon is LLM-TTA not the best in 2/3 OOD tasks, though we also note that only one TTA method didn't degrade Falcon's performance. OOD performance is our main focus in this work. For the ID datasets (Table 2), we find that LLM-TTA is the best approach for BERT and T5 in 2/3 ID datasets. In total, LLM-TTA is the best technique for BERT and T5 in 5/6 of the experiments for each of these models (Table 1 & 2). The Limitations section addresses TTA's decreasing effectiveness as task model size increases. We highlight our modest performance gains as evidence that, while TTA with a powerful augmentation function can improve performance in black-box settings, more work is needed to fully undo the performance regressions caused by OOD shifts (Section 5.1, 6, 7).
>
> **When is it more practical to use the LLM directly as the classifier?**
>
> Thank you for suggesting we further highlight comparing LLM-TTA versus using the LLM directly for the given task. To address your Weakness 3 and requested changes, we moved our existing analysis from Appendix A.7 to the main paper and added finer-grained results (see new Section 5.3). This experiment shows that pairing task-specific models with LLM augmentations is often better than using the augmentation LLM directly for the task. Specifically, for BERT and T5, LLM-TTA outperforms only using the LLM for 2/3 OOD tasks and 3/3 ID tasks. The LLM outperforms Falcon across all tasks. LLM-TTA often improves performance even when the LLM underperforms the task model. Beyond expanding our discussion of these results, we clarify that there can be cases where using the LLM directly is more practical (Sections 5.3 and 6).

---

### Author Response · Authors · 2024-05-09
**General Response**

We thank the reviewers for their valuable time and suggestions, which have helped improve the clarity and rigor of our analysis. We are glad that our technique's simplicity is a strength (LfbM, M6U3), that it is well motivated (LfbM, M6U3), and that our experiments provide a rigorous understanding of how well LLM-TTA works across common text classification tasks (aT3v, LfbM, M6U3).
We respond to the concerns shared by the reviewers and detail the changes we have made below.

**Should we study other tasks beyond text classification?**

Reviewers M6U3 and LfbM note that we focus on text classification and suggest that it needs to be clarified whether results on this task will extend to other domains, such as reasoning and generative tasks. We agree that extensions are possible, but we leave them to future work as we believe our work already provides a meaningful and comprehensive study of text classification, which is an important problem. We clarify the scope of this work in the Abstract as improving the robustness of black-box text classifiers. We have also included a paragraph in the limitations sections that acknowledges that our results on text classification may not generalize to other types of NLP problems. We highlight studying LLM-TTA in other settings as a direction for future work.

**Updates to the Manuscript**

1. We have run our OOD experiments multiple times with non-deterministic augmentation functions and report the mean accuracy with confidence intervals. The maximum standard deviation in accuracy for LLM-TTA is 0.05 percentage points. These new results show that there is a relatively low variance in performance across augmentation batches, suggesting that LLM-TTA’s performance is consistent across runs.
2. We added a section comparing ICR against using the augmentation LLM directly on the task. We find that the task model combined with ICR outperforms the LLM on all ID datasets and ⅔ OOD shifts. These results suggest that LLM-TTA can still improve performance even when the LLM underperforms the classifier on the task. However, there are some instances where using the LLM directly may be more practical. We clarify this point in the Limitations section as well.
3. Our Limitations section has been expanded to address some of the concerns raised by the reviewers.

---

### Decision · Action_Editor_Fhgt · 2024-06-18

**Recommendation:** Accept with minor revision

**Comment:**

This paper presents a new approach to improving the performance of text classifiers through the use of test-time data augmentation via an LLM. The main application is out-of-distribution classification, where the target text does not necessarily match the distribution that the classification model is fit to.

Two strategies are introduced: paraphrasing and in-context rewriting. Paraphrasing is a zero-shot approach that generates a diverse set of semantically equivalent inputs to the test input, whereas in-context rewriting uses a few-shot approach to rewrite the input in a similar style to provided examples. A strategy called entropy-based augmentation is introduced to select when to use the LLM for augmentation, saving time and cost while still yielding the benefits of test-time augmentation.

A few questions and concerns were raised by the reviewers around the novelty, scope, error bars, and comparisons to a fully in-context LLM. These were satisfactorily answered.

The remaining concerns raised by the reviewers include:
A request around a baseline that trains BERT or T5 from scratch (as opposed to fine-tuning). The statements made by the authors in their response (they are trained from scratch), and the text in the paper (they are fine-tuned, section 4.2) do not seem to agree.
A disagreement on the definition of OOD. There is no controllable, or quantifiable measure of OOD used in this paper, and one reviewer believes that the writing should shift focus towards generalization rather than OOD.

On this last point I would tend to agree. The term OOD is being used to mean “something that comes from a different distribution than the training set” without really specifying what is different about this distribution, or how different it is. Perhaps they are in-distribution, but simply challenging examples?

The reviewer suggested modifying the writing to focus less heavily on the concept of OOD. I would alternatively suggest writing a short section discussing this point, with some possible ideas on how to assess OODness in a measurable way in future work. The reviewer has already provided some suggestions around reasoning.

I think that with these changes, the paper satisfies the criteria for TMLR and should be accepted.

**Audience:**

Yes, there are many people interested in using LLMs for data augmentation.

**Claims And Evidence:**

Yes, the paper uses test-time augmentation and shows performance improvement on text classification tasks on OOD examples.

---

> ### Author Response · Authors · 2024-07-20
> **Response to Action Editor**
>
> Thank you for your valuable feedback and support of our work. We’re glad to see a positive reception and have taken the chance to further improve the quality of our manuscript. We detail how we’ve incorporated your feedback below:
>
> **Clarification on OOD vs. Generalization Narrative**
>
> Thank you for this suggestion. Our work indeed uses a broad notion of OOD, and our targeted contribution is not a comprehensive study of why models struggle to generalize to the evaluation datasets. We have addressed this feedback with prominent modifications to the Introduction, Experiment Setup, and Limitations sections. We also clarify that our “OOD” evaluation sets could indeed contain challenging ID examples. We discuss this point most in Limitations, where we cite alternative approaches and suggestions for how future work can more precisely measure how shifting distribution properties affect LLM-TTA.
>
> **Clarification on Fine-tuning vs. Training from Scratch**
>
> Thank you for raising this inconsistency. We mistakenly stated we had trained BERT and T5  from scratch in our response to Reviewer LfbM10. To be correct, we fine-tuned the popular public BERT and T5 models. Conducting in-house pretraining would indeed be an interesting additional experiment. However, it is unclear whether a model trained from scratch would generalize to the more popular and realistic setting of fine-tuning a strong pretrained model. We have clarified this point by adding a paragraph to the Limitations section acknowledging that the model's pretraining regimes can have a confounding factor in measuring LLM-TTA performance, particularly test-set leakage.